# Cascade Adversarial Machine Learning Regularized with a Unified Embedding

**Taesik Na, Jong Hwan Ko & Saibal Mukhopadhyay**
School of Electrical and Computer Engineering
Georgia Institute of Technology
Atlanta, GA 30332, USA
`{taesik.na, jonghwan.ko, smukhopadhyay6}@gatech.edu`

## Abstract

Injecting adversarial examples during training, known as adversarial training, can improve robustness against one-step attacks, but not for *unknown* iterative attacks. To address this challenge, we first show iteratively generated adversarial images easily transfer between networks trained with the *same strategy*. Inspired by this observation, we propose *cascade adversarial training*, which transfers the knowledge of the end results of adversarial training. We train a network from scratch by injecting iteratively generated adversarial images crafted from *already defended* networks in addition to one-step adversarial images from the network being trained. We also propose to utilize embedding space for both classification and low-level (pixel-level) similarity learning to ignore *unknown* pixel level perturbation. During training, we inject adversarial images *without replacing* their corresponding clean images and penalize the distance between the two embeddings (clean and adversarial). Experimental results show that cascade adversarial training together with our proposed low-level similarity learning efficiently enhances the robustness against iterative attacks, but at the expense of decreased robustness against one-step attacks. We show that combining those two techniques can also improve robustness under the *worst case* black box attack scenario.

## 1 Introduction

Injecting adversarial examples during training (adversarial training), (Goodfellow et al., 2015; Kurakin et al., 2017; Huang et al., 2015) increases the robustness of a network against adversarial attacks. The networks trained with one-step methods have shown noticeable robustness against one-step attacks, but, limited robustness against iterative attacks at test time. To address this challenge, we have made the following contributions:

**Cascade adversarial training:** We first show that iteratively generated adversarial images transfer well between networks when the source and the target networks are trained with the *same training method*. Inspired by this observation, we propose *cascade adversarial training* which transfers the knowledge of the end results of adversarial training. In particular, we train a network by injecting iter_FGSM images (section 2.1) crafted from an already *defended* network (a network trained with adversarial training) in addition to the one-step adversarial images crafted from the network being trained. The concept of using already trained networks for adversarial training is also introduced in (Tramèr et al., 2017). In their work, purely trained networks are used as another source networks to generate one-step adversarial examples for training. On the contrary, our cascade adversarial training uses already *defended* network for iter_FGSM images generation.

**Low level similarity learning:** We advance the previous data augmentation approach (Kurakin et al., 2017) by adding additional regularization in deep features to encourage a network to be insensitive to adversarial perturbation. In particular, we inject adversarial images in the mini batch *without replacing* their corresponding clean images and penalize distance between embeddings from the clean and the adversarial examples. There are past examples of using embedding space for learning similarity of high level features like face similarity between two different images (Schroff et al., 2015; Parkhi et al., 2015; Wen et al., 2016). Instead, we use the embedding space for learning

similarity of the pixel level differences between two similar images. The intuition of using this regularization is that *small difference on input should not drastically change the high level feature representation*.

**Analysis of adversarial training:**  We train ResNet models (He et al., 2016) on MNIST (LeCun & Cortes, 2010) and CIFAR10 dataset (Krizhevsky, 2009) using the proposed adversarial training. We first show low level similarity learning improves robustness of the network against adversarial images generated by *one-step* and *iterative* methods compared to the prior work. We show that modifying the weight of the distance measure in the loss function can help control trade-off between accuracies for the clean and adversarial examples.

Together with cascade adversarial training and low-level similarity learning, we achieve accuracy increase against unknown iterative attacks, but at the expense of decreased accuracy for one-step attacks. We also show our *cascade adversarial training and low level similarity learning* provide much better robustness against black box attack.

## 2    BACKGROUND ON ADVERSARIAL ATTACKS

### 2.1    ATTACK METHODS

**One-step fast gradient sign method (FGSM)**, referred to as "step_FGSM", generates adversarial image $\boldsymbol{X}^{adv}$ by adding sign of the gradients w.r.t. the clean image $\boldsymbol{X}$ multiplied by $\epsilon \in [0, 255]$ as shown below (Goodfellow et al., 2015):

$$\boldsymbol{X}^{adv} = \boldsymbol{X} + \epsilon \operatorname{sign}(\nabla_X J(\boldsymbol{X}, y_{true}))$$

**One-step target class method** generates $\boldsymbol{X}^{adv}$ by subtracting sign of the gradients computed on a target false label as follows:

$$\boldsymbol{X}^{adv} = \boldsymbol{X} - \epsilon \operatorname{sign}(\nabla_X J(\boldsymbol{X}, y_{target}))$$

We use least likely class $y_{LL}$ as a target class and refer this method as "step_ll".

**Basic iterative method**, referred to as "iter_FGSM", applies FGSM with small $\alpha$ multiple times.

$$\boldsymbol{X}_0^{adv} = \boldsymbol{X}, \quad \boldsymbol{X}_N^{adv} = Clip_{X,\epsilon}\big\{\boldsymbol{X}_{N-1}^{adv} + \alpha \operatorname{sign}(\nabla_{X_{N-1}^{adv}} J(\boldsymbol{X}_{N-1}^{adv}, y_{true}))\big\}$$

We use $\alpha = 1$, number of iterations $N$ to be $\min(\epsilon + 4, 1.25\epsilon)$. $Clip_{X,\epsilon}$ is elementwise clipping function where the input is clipped to the range $[\max(0, X - \epsilon), \min(255, X + \epsilon)]$.

**Iterative least-likely class method**, referred to as "iter_ll", is to apply "step_ll" with small $\alpha$ multiple times.

$$\boldsymbol{X}_0^{adv} = \boldsymbol{X}, \quad \boldsymbol{X}_N^{adv} = Clip_{X,\epsilon}\big\{\boldsymbol{X}_{N-1}^{adv} - \alpha \operatorname{sign}(\nabla_{X_{N-1}^{adv}} J(\boldsymbol{X}_{N-1}^{adv}, y_{LL}))\big\}$$

**Carlini and Wagner attack (Carlini & Wagner, 2017)** referred to as "CW" solves an optimization problem which minimizes both an objective function $f$ (such that attack is success if and only if $f(\boldsymbol{X}^{adv}) < 0$) and a distance measure between $\boldsymbol{X}^{adv}$ and $\boldsymbol{X}$.

**Black box attack** is performed by testing accuracy on a target network with the adversarial images crafted from a source network different from the target network. Lower accuracy means successful black-box attack. When we use the same network for both target and source network, we call this as white-box attack.

### 2.2    DEFENSE METHODS

**Adversarial training (Kurakin et al., 2017):**   is a form of data augmentation where it injects adversarial examples during training. In this method, $k$ examples are taken from the mini batch $B$ (size of $m$) and the adversarial examples are generated with one of step method. The $k$ adversarial examples *replaces* the corresponding clean examples when making mini batch. Below we refer this adversarial training method as "Kurakin's".

Table 1: CIFAR10 test results (%) under black box attacks for $\epsilon$=16. Source networks share the same initialization which is different from the target networks. {Target: R20, R20$_K$: standard, $K$urakin's, Source: R20$_2$, R20$_{K2}$: standard, $K$urakin's.}

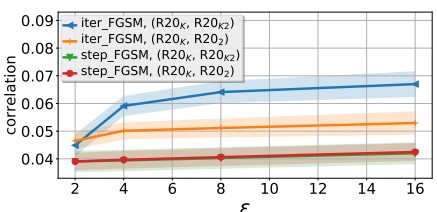

| Target | Source: step_FGSM | | Source: iter_FGSM | |
|---|---|---|---|---|
| | R20$_2$ | R20$_{K2}$ | R20$_2$ | R20$_{K2}$ |
| R20 | **16.2** | 31.6 | **2.7** | 60.1 |
| R20$_K$ | **66.7** | 82.7 | 55.8 | **28.5** |

Figure 1: Correlation between adversarial noises from different networks for each $\epsilon$. Shaded region shows $\pm$ 0.1 standard deviation of each line.

**Ensemble adversarial training (Tramèr et al., 2017):** is essentially the same with the adversarial training, but uses several pre-trained vanilla networks to generate one-step adversarial examples for training. Below we refer this adversarial training method as "Ensemble".

## 3 PROPOSED APPROACH

### 3.1 TRANSFERABILITY ANALYSIS

We first show transferability between purely trained networks and adversarially trained networks under black box attack. We use ResNet (He et al., 2016) models for CIFAR10 classification. We first train 20-layer ResNets with different methods (standard training, adversarial training (Kurakin et al., 2017)) and use those as target networks. We re-train networks (standard training and adversarial training) with the different initialization from the target networks, and use the trained networks as source networks. Experimental details and model descriptions can be found in Appendix A and B. In table 1, we report test accuracies under black box attack.

**Transferability (step attack)**: We first observe that high robustness against one-step attack between defended networks (R20$_{K2}$ -> R20$_K$), and low robustness between undefended networks (R20$_2$ -> R20). This observation shows that error surfaces of neural networks are *driven by the training method* and networks trained with the same method end up similar optimum states.

It is noteworthy to observe that the accuracies against step attack from the undefended network (R20$_2$) are always lower than those from defended network (R20$_{K2}$). Possible explanation for this would be that adversarial training tweaks gradient seen from the clean image to point toward weaker adversarial point along that gradient direction. As a result, one-step adversarial images from defended networks become weaker than those from undefended network.

**Transferability (iterative attack)**: We observe "iter_FGSM" attack remains very strong even under the black box attack scenario but *only between undefended networks or defended networks*. This is because iter_FGSM noises ($X^{adv}$-$X$) from defended networks resemble each other. As shown in figure 1, we observe higher correlation between iter_FGSM noises from a defended network (R20$_K$) and those from another defended network (R20$_{K2}$).

**Difficulty of defense/attack under the black box attack scenario**: As seen from this observation, it is efficient to attack an undefended/defended network with iter_FGSM examples crafted from another undefended/defended network. Thus, when we want to build a robust network under the black box attack scenario, it is desired to check accuracies for the adversarial examples crafted from other networks trained with the *same strategy*.

### 3.2 CASCADE ADVERSARIAL TRAINING

Inspired by the observation that iter_FGSM images transfer well between defended networks, we propose *cascade adversarial training*, which trains a network by injecting iter_FGSM images crafted from an already defended network. We hypothesize that the network being trained with cascade adversarial training will learn to avoid such adversarial perturbation, enhancing robustness against iter_FGSM attack. The intuition behind this proposed method is that we *transfer the knowledge of*

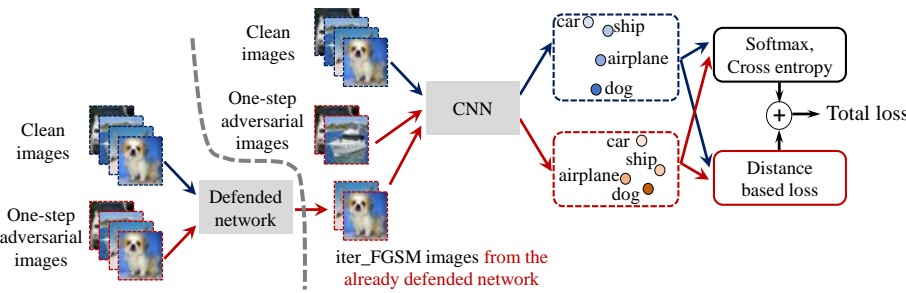

Figure 2: Cascade adversarial training regularized with a unified embedding.

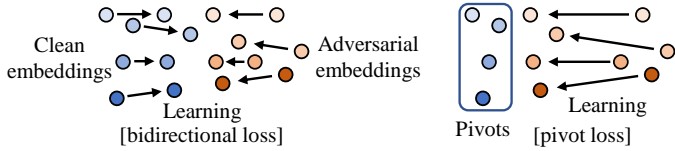

Figure 3: (**Left**) Bidirectional loss. (**Right**) Pivot loss.

*the end results of adversarial training*. In particular, we train a network by injecting iter_FGSM images crafted from already *defended* network in addition to the one-step adversarial images crafted from the network being trained.

### 3.3 REGULARIZATION WITH A UNIFIED EMBEDDING

We advance the algorithm proposed in (Kurakin et al., 2017) by adding low level similarity learning. Unlike (Kurakin et al., 2017), we include the clean examples used for generating adversarial images in the mini batch. Once one step forward pass is performed with the mini batch, embeddings are followed by the softmax layer for the cross entropy loss for the standard classification. At the same time, we take clean embeddings and adversarial embeddings, and minimize the distance between the two with the distance based loss.

The distance based loss encourages two similar images (clean and adversarial) to produce the same outputs, *not necessarily the true labels*. Thus, low-level similarity learning can be considered as an unsupervised learning. By adding regularization in higher embedding layer, convolution filters *gradually learn* how to ignore such pixel-level perturbation. We have applied regularization on lower layers with an assumption that low level pixel perturbation can be ignored in lower hierarchy of networks. However, adding regularization term on higher embedding layer right before the softmax layer showed best performance. *The more convolutional filters have chance to learn such similarity, the better the performance.* Note that cross entropy doesn't encourage two similar images to produce the same output labels. Standard image classification using cross entropy compares ground truth labels with outputs of a network *regardless of* how similar training images are.

The entire training process combining cascade adversarial training and low level similarity learning is shown in figure 2. We define the total loss as follows:

$$Loss = \frac{1}{(m-k)+\lambda k}\left(\sum_{i=1}^{m-k} L(\boldsymbol{X}_i|y_i) + \lambda\sum_{i=1}^{k} L(\boldsymbol{X}_i^{adv}|y_i)\right) + \lambda_2\sum_{i=1}^{k} L_{dist}(\boldsymbol{E}_i^{adv}, \boldsymbol{E}_i)$$

$\boldsymbol{E}_i$ and $\boldsymbol{E}_i^{adv}$ are the resulting embeddings from $\boldsymbol{X}_i$ and $\boldsymbol{X}_i^{adv}$, respectively. $m$ is the size of the mini batch, $k$ ($\leq m/2$) is the number of adversarial images in the mini batch. $\lambda$ is the parameter to control the relative weight of classification loss for adversarial images. $\lambda_2$ is the parameter to control the relative weight of the distance based loss $L_{dist}$ in the total loss.

**Bidirectional loss** minimizes the distance between the two embeddings by moving both clean and adversarial embeddings as shown in the left side of the figure 3.

$$L_{dist}(\boldsymbol{E}_i^{adv}, \boldsymbol{E}_i) = ||\boldsymbol{E}_i^{adv} - \boldsymbol{E}_i||_N^N, \quad N \in 1, 2 \quad i = 1, 2, ..., k$$

Table 2: MNIST test results (%) for 20-layer ResNet models ($\epsilon = 0.3*255$ at test time). { R20M: standard training, R20M$_K$: $K$urakin's adversarial training, R20M$_B$: $B$idirectional loss, R20M$_P$: $P$ivot loss.} CW $L_\infty$ attack is performed with 100 test samples (10 samples per each class) and the number of adversarial examples with $\epsilon > 0.3*255$ is reported. Additional details for CW attack can be found in Appendix F

| Model | clean | step_ll | step_FGSM | iter_ll | iter_FGSM | CW |
|---|---|---|---|---|---|---|
| R20M | 99.6 | 9.7 | 10.3 | 0.0 | 0.0 | 0 |
| R20M$_K$ | 99.6 | 96.7 | 94.5 | 89.0 | 60.2 | 46 |
| R20M$_B$ (Ours) | 99.5 | **97.3** | **96.2** | **97.2** | **88.5** | **81** |
| R20M$_P$ (Ours) | 99.5 | **97.1** | **95.7** | **96.9** | **88.9** | **82** |

We tried $N = 1, 2$ and found not much difference between the two. We report the results with $N = 2$ for the rest of the paper otherwise noted. When $N = 2$, $L_{dist}$ becomes L2 loss.

**Pivot loss** minimizes the distance between the two embeddings by moving only the adversarial embeddings as shown in the right side of the figure 3.

$$L_{dist}(\boldsymbol{E}_i^{adv}|\boldsymbol{E}_i) = ||\boldsymbol{E}_i^{adv} - \boldsymbol{E}_i||_N^N, \quad N \in 1, 2 \quad i = 1, 2, ..., k$$

In this case, clean embeddings ( $\boldsymbol{E}_i$ ) serve as pivots to the adversarial embeddings. In particular, we don't back-propagate through the clean embeddings for the distance based loss. The intuition behind the use of pivot loss is that the embedding from a clean image can be treated as the ground truth embedding.

## 4 LOW LEVEL SIMILARITY LEARNING ANALYSIS

### 4.1 EXPERIMENTAL RESULTS ON MNIST

We first analyze the effect of low level similarity learning on MNIST. We train ResNet models (He et al., 2016) with different methods (standard training, Kurakin's adversarial training and adversarial training with our distance based loss). Experimental details can be found in Appendix A.

Table 2 shows the accuracy results for MNIST test dataset for different types of attack methods. As shown in the table, our method achieves better accuracy than Kurakin's method for all types of attacks with a little sacrifice on the accuracy for the clean images. Even though adversarial training is done only with "step_ll", additional regularization increases robustness against *unknown* "step_FGSM", "iter_ll", "iter_FGSM" and CW $L_\infty$ attacks. This shows that our low-level similarity learning can successfully regularize the one-step adversarial perturbation and its vicinity for simple image classification like MNIST.

### 4.2 EMBEDDING SPACE VISUALIZATION

To visualize the embedding space, we modify 20-layer ResNet model where the last fully connected layer (64x10) is changed to two fully connected layers (64x2 and 2x10). We re-train networks with standard training, Kurakin's method and our pivot loss on MNIST. [1] In figure 4, we draw embeddings (dimension=2) between two fully connected layers. As seen from this figure, adversarial images from the network trained with standard training cross the decision boundary easily as $\epsilon$ increases. With Kurakin's adversarial training, the distances between clean and adversarial embeddings are minimized compared to standard training. And our pivot loss further minimizes distance between the clean and adversarial embeddings. Note that our pivot loss also decreases absolute value of the embeddings, thus, higher $\lambda_2$ will eventually result in overlap between distinct embedding distributions. We also observe that intra class variation of the clean embeddings are also minimized for the network trained with our pivot loss as shown in the scatter plot in figure 4 (c).

---

[1]Modified ResNet models showed slight decreased accuracy for both clean and adversarial images compared to original ResNet counterparts, however, we observed similar trends (improved accuracy for iterative attacks for the network trained with pivot loss) as in table 2.

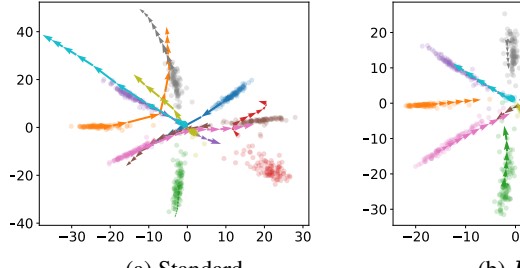

(a) Standard       (b) *K*urakin       (c) *P*ivot (Ours)

Figure 4: Embedding space visualization for modified ResNet models trained on MNIST. x-axis and y-axis show first and second dimension of embeddings respectively. Scatter plot shows first 100 clean embeddings per each class on MNIST test set. Each arrow shows difference between two embeddings (one from iter_FGSM image ($\epsilon$) and the other from ($\epsilon+8$)). We draw arrows from $\epsilon = 0$ to $\epsilon = 76$ ($\approx 0.3*255$) for one sample image per each class. We observe differences between clean and corresponding adversarial embeddings are minimized for the network trained with pivot loss.

## 4.3 EFFECT OF $\lambda_2$ ON CIFAR10

We train 20-layer ResNet models with pivot loss and various $\lambda_2$s for CIFAR10 dataset to study effects of the weight of the distance measure in the loss function. Figure 5 shows that a higher $\lambda_2$ increases accuracy of the iteratively generated adversarial images. However, it reduces accuracy on the clean images, and increasing $\lambda_2$ above 0.3 even results in divergence of the training. This is because embedding distributions of different classes will eventually overlap since absolute value of the embedding will be decreased as $\lambda_2$ increases as seen from the section 4.2. In this experiment, we show that there exists clear trade-off between accuracy for the clean images and that for the adversarial images, and we recommend using a very high $\lambda_2$ only under strong adversarial environment.

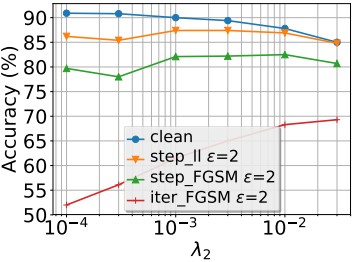

Figure 5: Accuracy vs. $\lambda_2$

## 5 CASCADE ADVERSARIAL TRAINING ANALYSIS

### 5.1 SOURCE NETWORK SELECTION

We further study the transferability of iter_FGSM images between various architectures. To this end, we first train 56-layer ResNet networks (Kurakin's, pivot loss) with the same initialization. Then we train another 56-layer ResNet network (Kurakin's) with different initialization. We repeat the training for the 110-layer ResNet networks. We measure correlation between iter_FGSM noises from different networks.

Figure 6 (a) shows correlation between iter_FGSM noises crafted from Kurakin's network and those from Pivot network with the same initialization. Conjectured from (Kurakin et al., 2017), we observe high corre-

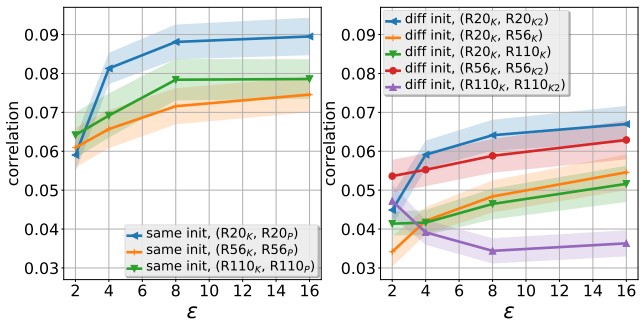

(a) Corr: same initialization      (b) Corr: different initialization

Figure 6: Correlation between iter_FGSM noises crafted from different networks for each $\epsilon$. Correlation is averaged over randomly chosen 128 images from CIFAR10 test-set.

Table 3: CIFAR10 test results (%) for 110-layer ResNet models. CW $L_\infty$ attack is performed with 100 test samples (10 samples per each class) and the number of adversarial examples with $\epsilon > 2$ or 4 is reported. {R110$_K$: $K$urakin's, R110$_P$: $P$ivot loss, R110$_E$: $E$nsemble training, R110$_{K,C}$: $K$urakin's and $C$ascade training, R110$_{P,E}$: $P$ivot loss and $E$nsemble training, and R110$_{P,C}$: $P$ivot loss and $C$ascade training}

| Model | clean | step_ll | | step_FGSM | | iter_FGSM | | CW | |
|---|---|---|---|---|---|---|---|---|---|
| | | $\epsilon$=2 | $\epsilon$=16 | $\epsilon$=2 | $\epsilon$=16 | $\epsilon$=2 | $\epsilon$=4 | $\epsilon$=2 | $\epsilon$=4 |
| R110$_K$ | 92.3 | **88.3** | **90.7** | **86.0** | **95.2** | 59.4 | 9.2 | 25 | 4 |
| R110$_P$ (Ours) | 92.3 | 86.0 | 89.4 | 81.6 | 91.6 | 64.1 | 20.9 | 32 | 7 |
| R110$_E$ | 92.3 | 86.3 | 74.3 | 84.1 | 72.9 | 63.5 | 21.1 | 24 | 6 |
| R110$_{K,C}$ (Ours) | 92.3 | 86.2 | 72.8 | 82.6 | 66.7 | 69.3 | 33.4 | 20 | 5 |
| R110$_{P,E}$ (Ours) | 91.3 | 84.0 | 65.7 | 77.6 | 54.5 | 66.8 | 38.3 | **38** | **16** |
| R110$_{P,C}$ (Ours) | 91.5 | 85.7 | 76.4 | 82.4 | 69.1 | **73.5** | **42.5** | 27 | 15 |

lation between iter_FGSM noises from networks with the same initialization. Correlation between iter_FGSM noises from the networks with different initialization, however, becomes lower as the network is deeper as shown in figure 6 (b). Since the degree of freedom increases as the network size increases, adversarially trained networks prone to end up with different states, thus, making transfer rate lower. To maximize the benefit of the cascade adversarial training, we propose to use *the same initialization for a cascade network and a source network* used for iterative adversarial examples generation.

## 5.2  WHITE BOX ATTACK ANALYSIS

We first compare a network trained with Kurakin's method and that with pivot loss. We train 110-layer ResNet models with/without pivot loss and report accuracy in table 3. We observe our low-level similarity learning further improves robustness against iterative attacks compared to Kurakin's adversarial training. However, the accuracy improvements against iterative attacks (iter_FGSM, CW) are limited, showing regularization effect of low-level similarity learning is not sufficient for the iterative attacks on complex color images like CIFAR10. This is different from MNIST test cases where we observed significant accuracy increase for iterative attacks only with pivot loss. We observe label leaking phenomenon reported in (Kurakin et al., 2017) happens even though we don't train a network with step_FGSM images. Additional analysis for this phenomenon is explained in Appendix D.

Next, we train a network from scratch with iter_FGSM examples crafted from the defended network, R110$_P$. We use the same initialization used in R110$_P$ as discussed in 5.1. In particular, iter_FGSM images are crafted from R110$_P$ with CIFAR10 training images for $\epsilon$= 1,2, ..., 16, and those are used randomly together with step_ll examples from the network being trained. We train cascade networks with/without pivot loss. We also train networks with ensemble adversarial training (Tramèr et al., 2017) with/without pivot loss for comparison. The implementation details for the trained models can be found in Appendix B.

We find several meaningful observations in table 3. First, ensemble and cascade models show improved accuracy against iterative attack although at the expense of decreased accuracy for one-step attacks compared to the baseline defended network (R110$_K$). Additional data augmentation from other networks enhances the robustness against iterative attack, weakening label leaking effect caused by one-step adversarial training.

Second, our low-level similarity learning (R110$_{P,E}$, R110$_{P,C}$) further enhances robustness against iterative attacks including fully unknown CW attack (especially for $\epsilon$=4). Additional knowledge learned from data augmentation through cascade/ensemble adversarial training enables networks to learn partial knowledge of perturbations generated by an iterative method. And the *learned iterative perturbations become regularized further with our low-level similarity learning* making networks robust against unknown iterative attacks.

Table 4: CIFAR10 test results (%) for 110-layer ResNet models under black box attacks ($\epsilon$=16). {Target: same networks in table 3 and R110$_K$: $\boldsymbol{K}$urakin's, Source: re-trained baseline, Kurakin's, cascade and ensemble networks with/without pivot loss. Source networks use the different initialization from the target networks. Additional details of the models can be found in Appendix B.}

| Target | Source: iter_FGSM | | | | | | |
|---|---|---|---|---|---|---|---|
| | R110$_2$ | R110$_{K2}$ | R110$_{E2}$ | R110$_{P2}$ | R110$_{K,C2}$ | R110$_{P,E2}$ | R110$_{P,C2}$ |
| R110$_K$ | 70.5 | 73.2 | **27.9** | 77.0 | 67.3 | 54.6 | 80.8 |
| R110$_E$ | 77.9 | 79.5 | 55.8 | 79.0 | 68.2 | **54.7** | 82.7 |
| R110$_P$ (Ours) | 75.9 | 75.6 | **39.6** | 78.5 | 68.3 | 61.3 | 83.3 |
| R110$_{K,C}$ (Ours) | **56.4** | 80.2 | 61.1 | 79.5 | 67.4 | 62.6 | 82.1 |
| R110$_{P,E}$ (Ours) | 78.2 | 82.1 | **67.7** | 81.7 | 73.4 | 68.4 | 83.8 |
| R110$_{P,C}$ (Ours) | 71.9 | 80.4 | **63.9** | 80.1 | 71.1 | 64.2 | 83.0 |

Third, our low-level similarity learning serves as a good regularizer for adversarial images, but not for the clean images for ensemble/cascade models (reduced accuracy for the clean images for R110$_{P,E}$ and R110$_{P,C}$ in table 3). Compared to pure adversarial training with one-step adversarial examples, the network sees more various adversarial perturbations during training as a result of ensemble and cascade training. Those perturbations are prone to end up embeddings in the vicinity of decision boundary more often than perturbations caused by one-step adversarial training. Pivot loss pulls the vicinity of those adversarial embeddings toward their corresponding clean embeddings. During this process, *clean embeddings from other classes* might also be moved toward the decision boundary which results in decreased accuracy for the clean images.

## 5.3 BLACK BOX ATTACK ANALYSIS

We finally perform black box attack analysis for the cascade/ensemble networks with/without pivot loss. We report black box attack accuracy with the source networks trained with the same method, but with different initialization from the target networks. The reason for this is adversarial examples transfer well between networks trained with the same strategy as observed in section 3.1. We re-train 110-layer ResNet models using Kurakin's, cascade and ensemble adversarial training with/without low-level similarity learning and use those networks as source networks for black-box attacks. Baseline 110-layer ResNet model is also included as a source network. Target networks are the same networks used in table 3. We found iter_FGSM attack resulted in lower accuracy than step_FGSM attack, thus, report iter_FGSM attack only in table 4.

We first observe that iter_FGSM attack from ensemble models (R110$_{E2}$, R110$_{P,E2}$) is strong (results in lower accuracy) compared to that from any other trained networks. [2] Since ensemble models learn various perturbation during training, adversarial noises crafted from those networks might be more general for other networks making them transfer easily between defended networks.

Second, cascade adversarial training breaks chicken and egg problem. (In section 3.1, we found that it is efficient to use a defended network as a source network to attack another defended network.) Even though the transferability between defended networks is reduced for deeper networks, cascade network (R110$_{K,C}$) shows worst case performance against the attack not from a defended network, but from a purely trained network (R110$_2$). Possible solution to further improve the worst case robustness would be to use more than one network as source networks (including pure/defended networks) for iter_FGSM images generation for cascade adversarial training.

Third, ensemble/cascade networks together with our low-level similarity learning (R110$_{P,E}$, R110$_{P,C}$) show better worst case accuracy under black box attack scenario. This shows that enhancing robustness against iterative white box attack also improves robustness against iterative black box attack.

---

[2]We also observed this when we switch the source and the target networks. Additional details can be found in Appendix E.

## 6 CONCLUSION

We performed through transfer analysis and showed iter_FGSM images transfer easily between networks trained with the *same strategy*. We exploited this and proposed cascade adversarial training, a method to train a network with iter_FGSM adversarial images crafted from already defended networks. We also proposed adversarial training regularized with a unified embedding for classification and low- level similarity learning by penalizing distance between the clean and their corresponding adversarial embeddings. Combining those two techniques (low level similarity learning + cascade adversarial training) with deeper networks further improved robustness against iterative attacks for both white-box and black-box attacks.

However, there is still a gap between accuracy for the clean images and that for the adversarial images. Improving robustness against both one-step and iterative attacks still remains challenging since it is shown to be difficult to train networks robust for both one-step and iterative attacks simultaneously. Future research is necessary to further improve the robustness against iterative attack *without* sacrificing the accuracy for step attacks or clean images under both white-box attack and black-box attack scenarios.

### ACKNOWLEDGMENTS

We thank Alexey Kurakin of Google Brain and Li Chen of Intel for comments that greatly improved the manuscript. The research reported here was supported in part by the Defense Advanced Research Projects Agency (DARPA) under contract number HR0011-17-2-0045. The views and conclusions contained herein are those of the authors and should not be interpreted as necessarily representing the official policies or endorsements, either expressed or implied, of DARPA.

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

## A    EXPERIMENTAL SETUP

We scale down the image values to [0,1] and don't perform any data augmentation for MNIST. For CIFAR10, we scale down the image values to [0,1] and subtract per-pixel mean values. We perform 24x24 random crop and random flip on 32x32 original images. We generate adversarial images with "step_ll" after these steps otherwise noted.

We use stochastic gradient descent (SGD) optimizer with momentum of 0.9, weight decay of 0.0001 and mini batch size of 128. For adversarial training, we generate $k = 64$ adversarial examples among 128 images in one mini-batch. We start with a learning rate of 0.1, divide it by 10 at 4k and 6k iterations, and terminate training at 8k iterations for MNIST, and 48k and 72k iterations, and terminate training at 94k iterations for CIFAR10. [3]

We also found that initialization affects the training results slightly as in (Kurakin et al., 2017), thus, we pre-train the networks 2 and 10 epochs for MNIST and CIFAR10, and use these as initial starting points for different configurations. We use $max\_e$ = 0.3*255 and 16 for MNIST and CIFAR10 respectively.

## B    MODEL DESCRIPTIONS

We summarize the model names used in this paper in table 5. For ensemble adversarial training, pre-trained networks as in table 7 together with the network being trained are used to generate one-step adversarial examples during training. For cascade adversarial training, pre-trained defended networks as in table 6 are used to generate iter_FGSM images, and the network being trained is used to generate one-step adversarial examples during training.

---

[3]We found that the adversarial training requires longer training time than the standard training. Authors in the original paper (He et al., 2016) changed the learning rate at 32k and 48k iterations and terminated training at 64k iterations.

Table 5: Model descriptions

| Dataset | ResNet | Initialization Group | Training | Model |
|---|---|---|---|---|
| MNIST | 20-layer | A | standard training | R20M |
| | | | $K$urakin's | R20M$_K$ |
| | | | $B$idirection loss | R20M$_B$ |
| | | | $P$ivot loss | R20M$_P$ |
| CIFAR10 | 20-layer | B | standard training | R20 |
| | | | $K$urakin's | R20$_K$ |
| | | | $E$nsemble training | R20$_E$ |
| | | | $B$idirection loss | R20$_B$ |
| | | | $P$ivot loss | R20$_P$ |
| | | | $K$urakin's & $C$ascade training | R20$_{K,C}$ |
| | | | $P$ivot loss & $E$nsemble training | R20$_{P,E}$ |
| | | | $P$ivot loss & $C$ascade training | R20$_{P,C}$ |
| | | C | standard training | R20$_2$ |
| | | | $K$urakin's | R20$_{K2}$ |
| | | | $P$ivot loss | R20$_{P2}$ |
| | | D | standard training | R20$_3$ |
| | | E | standard training | R20$_4$ |
| | 56-layer | F | $K$urakin's | R56$_K$ |
| | | | $P$ivot loss | R56$_P$ |
| | | G | $K$urakin's | R56$_{K2}$ |
| | 110-layer | H | standard training | R110 |
| | | | $K$urakin's | R110$_K$ |
| | | | $P$ivot loss | R110$_P$ |
| | | | $E$nsemble training | R110$_E$ |
| | | | $K$urakin's & $C$ascade training | R110$_{K,C}$ |
| | | | $P$ivot loss & $E$nsemble training | R110$_{P,E}$ |
| | | | $P$ivot loss & $C$ascade training | R110$_{P,C}$ |
| | | I | standard training | R110$_2$ |
| | | | $K$urakin's | R110$_{K2}$ |
| | | | $E$nsemble training | R110$_{E2}$ |
| | | | $P$ivot loss | R110$_{P2}$ |
| | | | $K$urakin's & $C$ascade training | R110$_{K,C2}$ |
| | | | $P$ivot loss & $E$nsemble training | R110$_{P,E2}$ |
| | | | $P$ivot loss & $C$ascade training | R110$_{P,C2}$ |
| | | J | standard training | R110$_3$ |
| | | K | standard training | R110$_4$ |

Table 6: Ensemble model description

| Ensemble models | Pre-trained models |
|---|---|
| R20$_E$, R20$_{P,E}$, R110$_E$, R110$_{P,E}$ | R20$_3$, R110$_3$ |
| R110$_{E2}$, R110$_{P,E2}$ | R20$_4$, R110$_4$ |

Table 7: Cascade model description

| Cascade models | Pre-trained model |
|---|---|
| R20$_{K,C}$, R20$_{P,C}$ | R20$_P$ |
| R110$_{K,C}$, R110$_{P,C}$ | R110$_P$ |
| R110$_{K,C2}$, R110$_{P,C2}$ | R110$_{P2}$ |

# C    ALTERNATIVE VISUALIZATION ON EMBEDDINGS

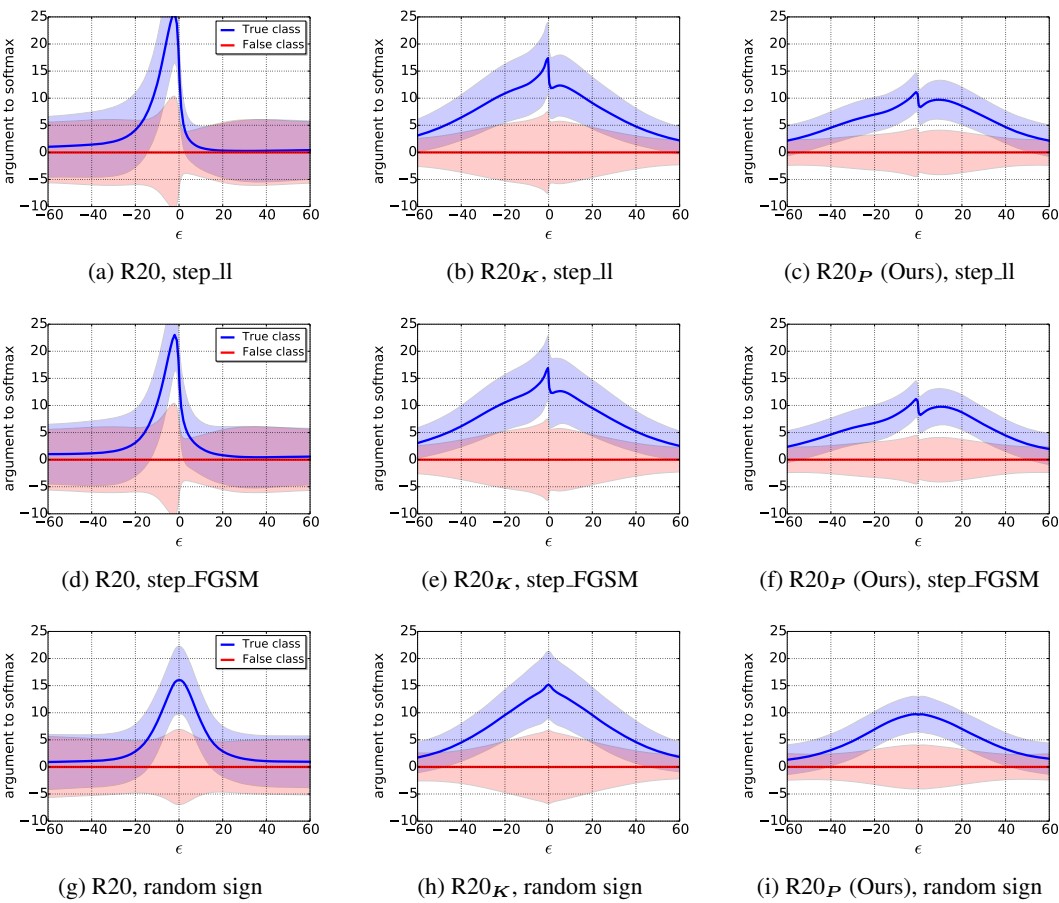

Figure 7: Argument to the softmax vs. $\epsilon$ in test time. "step_ll", "step_FGSM" and "random sign" methods were used to generate test-time adversarial images. Arguments to the softmax were measured by changing $\epsilon$ for each test method and averaged over randomly chosen 128 images from CIFAR10 test-set. Blue line represents true class and the red line represents mean of the false classes. Shaded region shows $\pm 1$ standard deviation of each line.

We draw average value of the argument to the softmax layer for the true class and the false classes to visualize how the adversarial training works as in figure 7. Standard training, as expected, shows dramatic drop in the values for the true class as we increase $\epsilon$ in "step_ll" or "step_FGSM direction. With adversarial training, we observe that the value drop is limited at small $\epsilon$ and our method even increases the value in certain range upto $\epsilon$=10. Note that adversarial training is not the same as the gradient masking. As illustrated in figure 7, it exposes gradient information, however, quickly distort gradients along the sign of the gradient ("step_ll" or "step_FGSM) direction. We also observe improved results (broader margins than baseline) for "random sign" added images even though we didn't inject random sign added images during training. Overall shape of the argument to the softmax layer in our case becomes smoother than Kurakin's method, suggesting our method is good for pixel level regularization. Even though actual value of the embeddings for the true class in our case is smaller than that in Kurakin's, the standard deviation of our case is less than Kurakin's, making better margin between the true class and false classes.

# D    LABEL LEAKING ANALYSIS

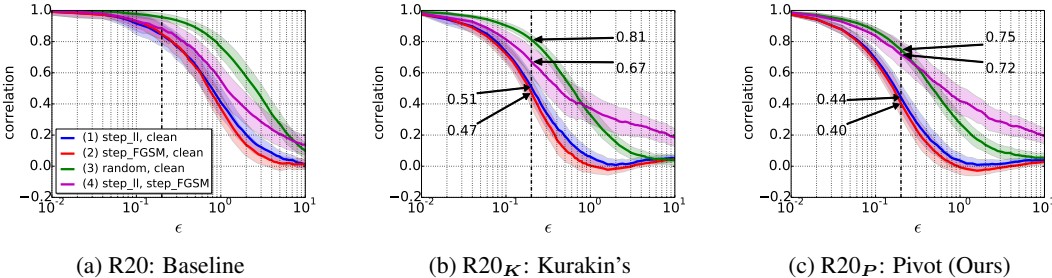

| (a) R20: Baseline | (b) R20$_K$: Kurakin's | (c) R20$_P$: Pivot (Ours) |

Figure 8: Averaged Pearson's correlation coefficient between the gradients w.r.t. two images. Correlation were measured by changing $\epsilon$ for each adversarial image and averaged over randomly chosen 128 images from CIFAR10 test-set. Shaded region represents $\pm\,0.5$ standard deviation of each line.

We observe accuracies for the "step_FGSM" adversarial images become higher than those for the clean images ("label leaking" phenomenon) by training with "step_FGSM" examples as in (Kurakin et al., 2017). Interestingly, we also observe "label leaking" phenomenon even without providing true labels for adversarial images generation. We argue that "label leaking" is a natural result of the adversarial training.

To understand the nature of adversarial training, we measure correlation between gradients w.r.t. different images (i.e. clean vs. adversarial) as a measure of error surface similarity. We measure correlation between gradients w.r.t. (1) clean vs. "step_ll" image, (2) clean vs. "step_FGSM" image, (3) clean vs. "random sign" added image, and (4) "step_ll" image vs. "step_FGSM" image for three trained networks (a) R20, (b) R20$_K$ and (c) R20$_P$ (Ours) in table 5. Figure 8 draws average value of correlations for each case.

**Meaning of the correlation:** In order to make strong adversarial images with "step_FGSM" method, correlation between the gradient w.r.t. the clean image and the gradient w.r.t. its corresponding adversarial image should remain high since the "step_FGSM" method *only* use the gradient w.r.t. the clean image ($\epsilon$=0). Lower correlation means perturbing the adversarial image at $\epsilon$ further to the gradient (seen from the clean image) direction is no longer efficient.

**Results of adversarial training:** We observe that (1) and (2) become quickly lower than (3) as $\epsilon$ increases. This means that, when we move toward the steepest (gradient) direction on the error surface, gradient is more quickly uncorrelated with the gradient w.r.t. the clean image than when we move to random direction. As a result of adversarial training, this uncorrelation is observed at a lower $\epsilon$ making one-step attack less efficient even with small perturbation. (1), (2) and (3) for our case are slightly lower than Kurakin's method at the same $\epsilon$ which means that our method is better at defending one-step attacks than Kurakin's.

**Error surface similarity between "step_ll" and "step_FGSM" images:** We also observe (4) remains high with higher $\epsilon$ for all trained networks. This means that the error surface (gradient) of the "step_ll" image and that of its corresponding "step_FGSM" image resemble each other. That is the reason why we get the robustness against "step_FGSM" method only by training with "step_ll" method and vice versa. (4) for our case is slightly higher than Kurakin's method at the same $\epsilon$ and that means our similarity learning tends to make error surfaces of the adversarial images with "step_ll" and "step_FGSM" method to be more similar.

**Analysis of label leaking phenomenon:** Interestingly, (2) becomes slightly negative in certain range ($1 < \epsilon < 3$ for Kurakin's, and $1 < \epsilon < 4$ for Pivot (Ours) ) and this could be the possible reason for "label leaking" phenomenon. For example, let's assume that we have a perturbed image (by "step_FGSM" method) at $\epsilon$ where the correlation between the gradients w.r.t. that image and the corresponding clean image is negative. Further increase of $\epsilon$ with the gradient (w.r.t. the clean image) direction actually decreases the loss resulting in increased accuracy (label leaking phenomenon). Due to the error surface similarity between "step_ll" and "step_FGSM" images and this negative

correlation effect, however, label leaking phenomenon can always happen for the networks trained with *one-step* adversarial examples.

# E   ADDITIONAL BLACK BOX ATTACK RESULTS

Table 8: CIFAR10 test results (%) under black box attacks between the network with the same initialization ($\epsilon$=16)}

| Target | Source: step_FGSM | | | Source: iter_FGSM | | |
|---|---|---|---|---|---|---|
| | R20 | $R20_K$ | $R20_P$ | R20 | $R20_K$ | $R20_P$ |
| R20 | **12.2** | 27.4 | 27.5 | 0.0 | 45.9 | 44.7 |
| $R20_K$ | **65.7** | 81.5 | 81.8 | 51.5 | 0.0 | **18.2** |
| $R20_P$ | **58.1** | 89.3 | 91.7 | 48.9 | **13.4** | 0.0 |

Table 8 shows that black box attack between trained networks with the same initialization tends to be more successful than that between networks with different initialization as explained in (Kurakin et al., 2017).

Table 9: CIFAR10 test results (%) under black box attacks for $\epsilon$=16. {Target and Source networks are switched from the table 1}

| Target | Source: step_FGSM | | | Source: iter_FGSM | | |
|---|---|---|---|---|---|---|
| | R20 | $R20_K$ | $R20_P$ | R20 | $R20_K$ | $R20_P$ |
| $R20_2$ | **17.9** | 33.9 | 34.5 | **4.1** | 54.8 | 54.3 |
| $R20_{K2}$ | **65.0** | 84.6 | 84.5 | 61.2 | **25.3** | **30.4** |
| $R20_{P2}$ | **66.4** | 88.2 | 87.2 | 61.6 | **27.7** | **36.1** |

In table 9, our method ($R20_{P2}$) is always better at one-step and iterative black box attack from defended networks ($R20_K$, $R20_P$) and undefended network (R20) than Kurakin's method ($R20_{B2}$). However, it is hard to tell which method is better than the other one as explained in the main paper.

Table 10: CIFAR10 test results (%) for cascade networks under black box attacks for $\epsilon$=16. {Target and Source: Please see the model descriptions in Appendix B.}

| Target | Source: iter_FGSM | | | | | | |
|---|---|---|---|---|---|---|---|
| | R110 | $R110_K$ | $R110_E$ | $R110_P$ | $R110_{K,C}$ | $R110_{P,E}$ | $R110_{P,C}$ |
| $R110_{K2}$ | 80.5 | 72.7 | 49.3 | 68.0 | 49.6 | **41.0** | 67.9 |
| $R110_{E2}$ | 82.7 | 59.1 | **39.5** | 59.6 | 51.5 | 40.3 | 69.6 |
| $R110_{P2}$ (Ours) | 80.3 | 75.9 | 54.2 | 72.2 | 54.9 | **44.3** | 72.4 |
| $R110_{K,C2}$ (Ours) | 62.1 | 74.7 | 61.5 | 72.3 | 46.5 | **39.0** | 67.9 |
| $R110_{P,E2}$ (Ours) | 81.5 | 79.0 | 50.0 | 77.3 | 56.9 | **45.5** | 75.2 |
| $R110_{P,C2}$ (Ours) | 72.2 | 76.4 | 60.6 | 73.8 | 51.0 | **40.9** | 72.0 |

In table 10, we show black box attack accuracies with the source and the target networks switched from the table 4. We also observe that networks trained with both low-level similarity learning and cascade/ensemble adversarial training ($R110_{P,C2}$, $R110_{P,E2}$) show better *worst-case* performance than other networks. Overall, iter_FGSM images crafted from ensemble model families ($R110_E$, $R110_{P,E}$) remain strong on the defended networks.

# F Implementation Details for Carlini-Wagner $L_\infty$ Attack

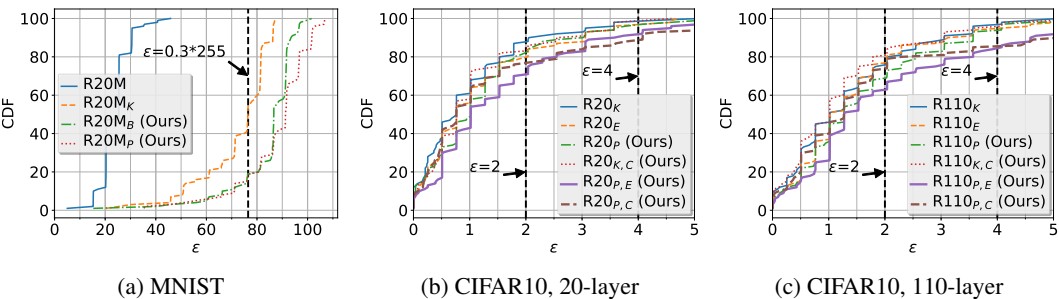

(a) MNIST        (b) CIFAR10, 20-layer        (c) CIFAR10, 110-layer

Figure 9: Cumulative distribution function vs. $\epsilon$ for 100 test adversarial examples generated by CW $L_\infty$ attack. Lower CDF value for a fixed $\epsilon$ means the better defense.

Carlini and Wagner (CW) $L_\infty$ attack solves the following optimization problem for every input $\boldsymbol{X}$.

$$minimize \quad c \cdot f(\boldsymbol{X} + \boldsymbol{\delta}) + \sum_i [(|\delta_i| - \tau)^+]$$

$$such\ that \quad \boldsymbol{X} + \boldsymbol{\delta} \in [0, 1]^n$$

where, the function $f$ is defined such that attack is success if and only if $f(\boldsymbol{X} + \boldsymbol{\delta}) < 0$, $\boldsymbol{\delta}$ is the target perturbation defined as $\boldsymbol{X}^{adv} - \boldsymbol{X}$, $c$ is the parameter to control the relative weight of function $f$ in the total cost function, and $\tau$ is the control threshold used to penalize any terms that exceed $\tau$.

Since CW $L_\infty$ attack is computationally expensive, we only use 100 test examples (10 examples per each class). We search adversarial example $\boldsymbol{X}^{adv}$ with $c \in \{0.1, 0.2, 0.5, 1, 2, 5, 10, 20\}$ and $\tau \in \{0.02, 0.04, ..., 0.6\}$ for MNIST and $c \in \{0.1, 0.3, 1, 3, 10, 30, 100\}$ and $\tau \in \{0.001, 0.002, ..., 0.01, 0.012, ..., 0.02, 0.024, ..., 0.04, 0.048, ..., 0.08\}$ for CIFAR10. We use Adam optimizer with an initial learning rate of $0.01/c$ since we found constant initial learning rate for $c \cdot f(\boldsymbol{X} + \boldsymbol{\delta})$ term is critical for successful adversarial images generation. We terminate the search after 2,000 iterations for each $\boldsymbol{X}$, $c$ and $\tau$. If $f(\boldsymbol{X} + \boldsymbol{\delta}) < 0$ and the resulting $||\boldsymbol{\delta}||_\infty$ is lower than the current best distance, we update $\boldsymbol{X}^{adv}$.

Figure 9 shows cumulative distribution function of $\epsilon$ for 100 successful adversarial examples per each network. We report the number of adversarial examples with $\epsilon > 0.3*255$ for MNIST and that with $\epsilon > 2$ or 4 for CIFAR10. As seen from this figure, our approaches provide robust defense against CW $L_\infty$ attack compared to other approaches.

