# OpenReview forum: "Cascade Adversarial Machine Learning Regularized with a Unified Embedding"
_ICLR.cc/2018/Conference — Accept (Poster)_

### Official Review · AnonReviewer1 · 2017-11-28

**Rating:** 6
**Confidence:** 4

**Review:**

The authors proposed to supplement adversarial training with an additional regularization that forces the embeddings of clean and adversarial inputs to be similar. The authors demonstrate on MNIST and CIFAR that the added regularization leads to more robustness to various kinds of attacks. The authors further propose to enhance the network with cascaded adversarial training, that is, learning against iteratively generated adversarial inputs, and showed improved performance against harder attacks.

The idea proposed is fairly straight-forward. Despite being a simple approach, the experimental results are quite promising.  The analysis on the gradient correlation coefficient and label leaking phenomenon provide some interesting insights.

As pointed out in section 4.2, increasing the regularization coefficient leads to degenerated embeddings. Have the authors consider distance metrics that are less sensitive to the magnitude of the embeddings, for example, normalizing the inputs before sending it to the bidirectional or pivot loss, or use cosine distance etc.?

Table 4 and 5 seem to suggest that cascaded adversarial learning have more negative impact on test set with one-step attacks than clean test set, which is a bit counter-intuitive. Do the authors have any insight on this?

Comments:
1. The writing of the paper could be improved. For example, "Transferability analysis" in section 1 is barely understandable;
2. Arrow in Figure 3 are not quite readable;
3. The paper is over 11 pages. The authors might want to consider shrink it down the recommended length.

---

> ### Author Response · Authors · 2017-12-25
> **Thanks for the feedback**
>
> Thank you for the valuable reviews.
>
> Q1 - As pointed out in section 4.2, increasing the regularization coefficient leads to degenerated embeddings. Have the authors consider distance metrics that are less sensitive to the magnitude of the embeddings, for example, normalizing the inputs before sending it to the bidirectional or pivot loss, or use cosine distance etc.?
> (Ans) We would like to mention that every regularization technique including weight decay (L1/L2 regularization) has the problem of degenerated embeddings if we weigh more on the regularized term. We have applied regularization after normalizing embeddings (divided by the standard deviation of the embeddings). As we increase lambda_2, the mean of the embeddings remains the same, but, the standard deviation of the embeddings becomes large. That means intra class variation becomes large. We eventually observed degenerated embeddings (lower accuracy on the clean example) for large lambda_2 which is the same phenomenon with our original version of implementation.
>
> Q2- Table 4 and 5 seem to suggest that cascaded adversarial learning have more negative impact on test set with one-step attacks than clean test set, which is a bit counter-intuitive. Do the authors have any insight on this?
> (Ans) The purpose of cascade adversarial training is to improve the robustness against “iterative” attack. And we showed the effectiveness of the method showing increased accuracy against iterative attack, but, at the expense of decreased accuracy against one-step attack.
> We have observed this phenomenon (the networks shown to be robust against iterative attacks tends to less robust against one-step attacks) in various conditions including ensemble adversarial training. We feel that in somehow, there is trade-off between them. Based on our extensive empirical experiments, it was very hard to increase robustness against both one-step attack and iterative attack. It is a still open question why this is so difficult, but, we assume that high dimensionality of the input is one reason for this. That means once we find good defense for some adversarial direction, there exists another adversarial direction which breaks the defense.
> Finally, we would like to mention that even though we have observed decreased accuracy for the one-step attack from cascade adversarial training, accuracy gain on the iterative attacks in white box setting helps increase in robustness against black box attack (for both one-step and iterative attacks.)
>
> Q3 - The writing of the paper could be improved. For example, "Transferability analysis" in section 1 is barely understandable;
> (Ans) We’ve updated the manuscript. Essentially, the detailed analysis can be found in new section 3.1.
>
> Q4 - Arrow in Figure 3 are not quite readable;
> (Ans) We’ve re-drawn the arrow from e to e+8 instead of e+4 to increase readability in revised version.
>
> Q5 - The paper is over 11 pages. The authors might want to consider shrink it down the recommended length.
> (Ans) We’ve updated the manuscript. Thanks for your recommendation.

---

### Official Review · AnonReviewer2 · 2017-11-29

**Rating:** 6
**Confidence:** 4

**Review:**

This paper improves adversarial training by adding to its traditional objective a regularization term forcing a clean example and its adversarial version to be close in the embedding space. This is an interesting idea which, from a robustness point of view (Xu et al, 2013) makes sense. Note that a similar strategy has been used in the recent past under the name of stability training. The proposed method works well on CIFAR and MNIST datasets. My main concerns are:

	- The adversarial objective and the stability objective are potentially conflicting. Indeed when the network misclassifies an example, its adversarial version is forced to be close to it in embedding space while the adversarial term promotes a different prediction from the clean version (that of the ground truth label). Have the authors considered this issue? Can they elaborate more on how they with this?

	- It may be significantly more difficult to make this work in such setting due to the dimensionality of the data. Did the authors try such experiment? It would be interesting to see these results.

Lastly, The insights regarding label leaking are not compelling.  Label leaking is not a mysterious phenomenon. An adversarially trained model learns on two different distributions. Given the fixed size of the hypothesis space explored (i.e., same architecture used for vanilla and adversarial training), It is natural that the statistics of the simpler distribution are captured better by the model. Overall, the paper contains valuable information and a method that can contribute to the quest of more robust models. I lean on accept side.

---

> ### Author Response · Authors · 2017-12-25
> **Thanks for the feedback**
>
> Thank you for the valuable reviews.
>
> Q1.- The adversarial objective and the stability objective are potentially conflicting. Indeed when the network misclassifies an example, its adversarial version is forced to be close to it in embedding space while the adversarial term promotes a different prediction from the clean version (that of the ground truth label). Have the authors considered this issue? Can they elaborate more on how they with this?
> (Ans) Yes, we totally agree with that the objective of the image classification and similarity objective can be potentially conflicting each other. We would like to address that our distance based loss can be considered as a way of regularization. Like every regularization, for example, weight L1/L2 regularization which penalizes the large value of the weights, too much regularization can always damage the training process.
> If the classifier misclassifies the clean version, that means the example is hard example. Adversarial version of the hard example will also be misclassified. Instead of trying to always encourage to produce ground truth, the similarity loss will encourage adversarial version to mimic the clean version. We can think this is somewhat analogous to student-teacher learning where student network is trained with the soft target from the teacher network instead of conventional hard target.
>
> Q2- It may be significantly more difficult to make this work in such setting due to the dimensionality of the data. Did the authors try such experiment? It would be interesting to see these results.
> (Ans) We have applied low level similarity learning for ImageNet dataset. We observed similar results (A network trained with pivot loss showed improved accuracy for white box iterative attacks compared to the network trained with adversarial training only.) We will augment those results in the final version. Due to the lack of computing resources, however, we haven’t tried training several similar networks with different initialization and testing those networks under white box and black box scenario.
> If you are talking about the dimensionality of the embedding, we actually have applied similarity loss on un-normalized logits (the layer right before the softmax layer) where the dimension of the embedding is exact the same with the number of labels which we don’t think a problem. We have tried applying similarity loss on intermediate neurons (that means different size of the embeddings), and found that applying similarity loss is efficient when we apply this at the very end of the network.
>
> Q3- Lastly, The insights regarding label leaking are not compelling
> (Ans) There is no clear evidence that the distribution of the one-step adversarial images is simpler than the distribution of the clean images. Adversarial images have meant to be created to fool the network. If we think the images with random noise, and train a network with clean and noisy images, we observe the decreased accuracy for noisy images since we lost the information due to the noise. Considering the fact that the adversarial images can be viewed as images with additive noise, label leaking phenomenon is not well understood since we actually added some noise which is intentional. Correlation analysis reveals the reason behind this effect.

---

### Official Review · AnonReviewer3 · 2017-11-29

**Rating:** 5
**Confidence:** 4

**Review:**

The paper presents a novel adversarial training setup, based on distance based loss of the feature embedding.

+ novel loss
+ good experimental evaluation
+ better performance
- way too long
- structure could be improved
- pivot loss seems hacky

The distance based loss is novel, and significantly different from prior work. It seems to perform well in practice as shown in the experimental section.
The experimental section is extensive, and offers new insights into both the presented algorithm and baselines. Judging the content of the paper alone, it should be accepted.

However, the exposition needs significant improvements to warrant acceptance. First, the paper is way too long and unfocused. The recommended length is 8 pages + 1 page for citations. This paper is 12+1 pages long, plus a 5 page supplement. I'd highly recommend the authors to cut a third of their text, it would help focus the paper on the actual message: pushing their new algorithm. Try to remove any sentence or word that doesn't serve a purpose (help sell the algorithm).
The structure of the paper could also be improved. For example the cascade adversarial training is buried deep inside the experimental section. Considering that it is part of the title, I would have expected a proper exposition of the idea in the technical section (before any results are presented). While condensing the paper, consider presenting all technical material before evaluation.
Finally, the pivot "loss" seems a bit hacky. First, the pivot objective and bidirectional loss are exactly the same thing. While the bidirectional loss is a proper loss and optimized as such (by optimizing both E^adv and E), the pivot objective is no loss function, as it does not correspond to any function any optimization algorithm could minimize. I'd recommend the just remove the pivot objective, or at least not call it a loss.

In summary, the results and presented method are good, and eventually deserve publication. However the exposition needs to significantly improve for the paper to be ready for ICLR.

---

> ### Author Response · Authors · 2017-12-25
> **Thanks for the feedback**
>
> Thank you for the valuable reviews.
>
> Q1 – Way too long, structure could be improved
> (Ans) Thanks for your feedback. We have changed the structure/length of the paper in revised version. In revised version, the contents are essentially the same with those in the previous version. We only changed structure/length of the paper. As the reviewer suggested, we have included exposition of the idea in the technical section before the experimental results.
> Only the transferability analysis has been included before the experimental results since we found it is necessary to show ‘higher transferability of the iterative adversarial examples between defended networks’ to introduce our proposed cascade adversarial training.
> Thank you very much for your valuable feedback which greatly improved the quality of the revised manuscript.
>
> Q2 - pivot loss seems hacky
> (Ans) Initially, we didn’t want to hurt the accuracy for the clean data, and that motivated us to invent pivot loss. In pivot loss, we assume clean embeddings as ground truth embeddings that adversarial embeddings have to mimic.
> Bidirectional loss and pivot loss have the same mathematical form, however, in pivot loss, embeddings from the clean examples are treated as constants (non-trainable). For pivot loss, it actually computes gradients with back-propagation, but only through embeddings computed from adversarial images.
> We can also think this is somewhat analogous to student-teacher learning where student network (adversarial embedding) is trained with the soft target from the teacher network (clean embedding) instead of conventional hard target.

---

### Author Response · Authors · 2017-12-25
**Summary of the revision**

We have updated manuscript to make it concise. The contents are essentially the same with the previous version. We only changed structure/length of the paper. As the reviewers suggested, we have included exposition of the idea in the technical section before the experimental results. Only the transferability analysis has been included before the experimental results since we found it is necessary to show ‘higher transferability of the iterative adversarial examples between defended networks’ to introduce our proposed cascade adversarial training. The table of contents are changed as follows.

1. introduction
2. Background on Adversarial Attacks
    2.1 attack methods
    2.2 defense methods
3. Proposed Approach
    3.1 Trasferability analysis
    3.2 Cascade adversarial training
    3.3 Regularization with a unified embedding
4. Low Level Similarity Learning Analysis
    4.1 Experimental results on MNIST
    4.2 Embedding space visualization
    4.3 Effect of lambda_2 on CIFAR10
5. Cascade Adversarial Training Analysis
    5.1 Source network selection
    5.2 White box attack results
    5.3 Black box attack results
6. Conclusion

Materials moved to Appendix
- experimental setup
- label leaking analysis (We think it gives fruitful insights into the adversarial training, however, we have decided to move this analysis to Appendix since it is not directly related with our proposed idea.)

Materials removed in the manuscript
- ResNet 20-layer white box attack results on CIFAR10

Additional comments (practical importance of low-level similarity loss)
- We did not touch on the importance of the low-level similarity learning enough in the main paper. However, we would like to emphasize the role of low-level similarity learning (regularization) for practical reason. This very simple technique can be used as a knob for controlling the trade off between accuracy for the clean and that for the adversarial inputs. Thus, one can train a network with high lambda_2 for enhanced robustness for critical applications like autonomous driving. And our method can be combined with any other orthogonal approaches (non-differentiable input transformation, feature squeezing) for further improved robustness.

Again, we would like to mention that the technical contents are exact the same with those in the previous version. We would like to kindly ask reviewers to re-evaluate the paper focusing more on the technical contribution of the paper. Thank you.

---

### Author Response · Authors · 2018-01-03
**Submission similar to our work in OpenReview**

We would like to mention that the submission similar to our work in OpenReview.

- Ensemble Adversarial Training: Attacks and Defenses (avg. score of 6.0)
https://openreview.net/forum?id=rkZvSe-RZ
This work is essentially the work we've used as a reference in our paper.
We showed that combining low-level similarity loss and ensemble adversarial training results in superior accuracy against adversarial attacks under both white/black box attacks compared to the vanilla ensemble adversarial training approach.

In table 3 (Accuracy under the white box attack)
Ensemble adversarial training 24% @ CW e=2
Ensemble adversarial training with pivot loss (Ours) 38% @ CW e=2

In table 4 (Worst case accuracy under the black box attack)
Ensemble adversarial training 54.7%
Ensemble adversarial training with pivot loss (Ours) 67.7%

---

### Decision · Program_Chairs · 2018-01-29
**ICLR 2018 Conference Acceptance Decision**

**Decision:**

Accept (Poster)

**Comment:**

This paper forms a good contribution to the active area of adversarial training.  The main issue with the original submission was presentation quality and excessive length.  The revised version is much improved.  However, it still needs some work on the writing, in large part in the transferability section but also to clean up a large number of non-native formulations like missing/extra determiners and some awkward phrasing.  It should be carefully proofread by a native English speaker if possible.  Also, the citation formatting is incorrect (frequently using \citet instead of \citep).